# Diagnosis of Hodgkin Lymphoma from Cell Block: A Reliable and Helpful Tool in “Selected” Diagnostic Practice

**DOI:** 10.3390/diagnostics10100748

**Published:** 2020-09-25

**Authors:** Paola Parente, Claudia Covelli, Magda Zanelli, Domenico Trombetta, Illuminato Carosi, Cristiano Carbonelli, Marco Sperandeo, Luca Mastracci, Giovanni Biancofiore, Maurizio Zizzo, Marco Taurchini, Stefano Ascani, Paolo Graziano

**Affiliations:** 1Pathology Unit, Fondazione IRCCS Casa Sollievo della Sofferenza, 71013 San Giovanni Rotondo, Italy; paolaparente77@gmail.com (P.P.); cla.covelli85@gmail.com (C.C.); i.carosi@operapadrepio.it (I.C.); gbiancofiore@libero.it (G.B.); p.graziano@operapadrepio.it (P.G.); 2Pathology Unit, Azienda Unità Sanitaria Locale-IRCCS di Reggio Emilia, 42123 Reggio Emilia, Italy; Magda.Zanelli@ausl.re.it; 3Laboratory of Oncology, Fondazione IRCCS Casa Sollievo della Sofferenza, 71013 San Giovanni Rotondo, Italy; 4Unit of Interventional Pulmonology, Department of Internal Medicine, Fondazione IRCCS Casa Sollievo della Sofferenza, 71013 San Giovanni Rotondo, Italy; c.carbonelli@operapadrepio.it; 5Unit of Interventional and Diagnostic Ultrasound of Internal Medicine, Fondazione IRCCS Casa Sollievo della Sofferenza, 71013 San Giovanni Rotondo, Italy; sperandeomar@gmail.com; 6Anatomic Pathology, Department of Surgical Sciences and Integrated Diagnostics (DISC), University of Genova, 16132 Genova, Italy; luca.mastracci@unige.it; 7Anatomic Pathology, Ospedale Policlinico San Martino IRCCS, 16132 Genova, Italy; 8Surgical Oncology Unit, Azienda Unità Sanitaria Locale-IRCCS di Reggio Emilia, 42123 Reggio Emilia, Italy; zizzomaurizio@gmail.com; 9Thoracic Surgical Unit, Fondazione IRCCS Casa Sollievo della Sofferenza, 71013 San Giovanni Rotondo, Italy; m.taurchini@operapadrepio.it; 10Pathology Unit, Hospital of Terni and University of Perugia, 05100 Terni, Italy; s.ascani@aospterni.it

**Keywords:** cell block, cytology, Hodgkin lymphoma, non-Hodgkin lymphoma, EBUS-FNA, US-FNA, paraffin block, tumor pathology

## Abstract

Background: The diagnosis of lymphoma requires surgical specimens to perform morphological evaluation, immunohistochemical and molecular analyses. Ultrasound-guided fine needle aspiration may represent an appropriate first approach to obtain cytological samples in impalpable lesions and/or in patients unsuitable for surgical procedures. Although cytology has intrinsic limitations, the cell block method may increase the possibility of achieving an accurate diagnosis. Methods: We retrospectively selected a total of 47 ultrasound-guided fine needle aspiration and drainage samples taken from patients with effusion and deep-seated lesions which are clinically suspicious in terms of malignancy. Results: In 27 cases, both cell block and conventional cytology were performed: 21/27 cell blocks were adequate for the diagnosis of lymphoma and suitable for immunocytochemistry and molecular analyses vs. 12/20 samples to which only conventional cytology was applied. Moreover, in five patients we were able to make a diagnosis of Hodgkin lymphoma with the cell block (CB) technique. Conclusions: Contrary to conventional cytology, the cell block method may allow immunocytochemistry and molecular studies providing useful information for the diagnosis and subtypization of lymphoma in patients unsuitable for surgical procedure or with deep-seated lesions or extra-nodal diseases; additionally, it is a daily, simple and helpful approach. Moreover, we describe the usefulness of cell blocks in the diagnosis of Hodgkin lymphoma.

## 1. Introduction

Lymphoma is a malignant neoplasm of the hematopoietic tissue. It includes different non-Hodgkin lymphoma (NHL) and Hodgkin lymphoma (HL) subtypes. According to the current World Health Organization (WHO) classification, an integrated diagnostic system based on morphological pattern recognition, immunophenotyping and molecular analyses is recommended in clinical practice to establish a correct diagnosis, provide prognostic information and choose the most appropriate and effective therapy [1,2]. Surgical biopsy is considered the gold standard for the diagnosis of lymphoma [3], for it evaluates the architectural pattern of growth, establishing the grading, as for follicular lymphoma [4], and identifies possible areas that may progress into more aggressive forms [5]. However, for selected patients unsuitable for surgical procedures presenting extra-nodal or deep-seated lesions (such as intrathoracic, intra-abdominal, or other anatomical sites where the lesions are impalpable or not easily accessible by superficial biopsy and for which radiological guidance is required for appropriate sampling) or with recurrent lymphoma and/or poor performance status, ultrasound-guided fine needle aspiration (US-FNA) and endobronchial ultrasound-guided fine needle aspiration (EBUS-FNA) may be considered as alternative techniques to obtain diagnostic samples [6,7]. It has been suggested that conventional cytology (CC), specifically smears and cytocentrifuge preparations, obtained from fine needle aspiration (FNA), combined with cytofluorimetry (CF) might be useful to achieve a diagnosis of NHL [1]. Nevertheless, CC does not allow the architecture of lymphoproliferative diseases to be recognized and has limitations due to its scattered cellularity and inability to perform immunocytochemical studies and Fluorescence In Situ Hybridization (FISH). HL can be only suspected with CC, but the final diagnosis, for which adequate neoplastic cellularity and immunocytochemical characterization are necessary, cannot be reached [8]. Moreover, CF does not allow the diagnosis of HL [6] and large B-cell lymphoma, with the latter generally having a fragile cytoplasm with a subsequent artificial loss of surface antigen [3,9]. Furthermore, in T-cell lymphomas, the T-cell receptor γ gene rearrangement clonality cannot be demonstrated by CF [10]. The Cell Block (CB) technique is one of the oldest methods for the evaluation of body cavity fluids and a well-established tool for the diagnostic and molecular profiling of pulmonary neoplasms [11,12]; in combination with CF, it also enables the diagnosis of NHL [1,8]. Adequate cellularity, partially preserved architecture and satisfactory details of the nucleus and cytoplasm might be obtained using CB; moreover, immunocytochemical and molecular analyses can also be performed on CBs [13].

We compared the final diagnoses in a retrospective series of 47 samples obtained from 36 US-FNAs, 3 EBUS-FNAs and 8 drainages in patients with effusion, deep-seated lesions or extra-nodal diseases with a clinical suspicion of lymphoma or carcinoma. We evaluated the possibility to achieve a conclusive diagnosis of NHL by CB, with the aid of immunophenotyping and FISH, and a first diagnosis of HL.

## 2. Materials and Methods

We searched in our electronic data system all consecutive FNAs (EBUS-FNAs, US-FNAs) and drainages performed on patients with deep-seated lesions or extra-nodal lesions with clinical suspicion of lymphoma (first diagnosis and/or relapsing disease) and with a final diagnosis of lymphoma from October 2013 to November 2019. Patient demographics, radiological findings, procedure details as well as follow-up data were collected from our Electronic Hospital Database (SISWEB). A total of 36 US-FNAs, 3 EBUS-FNAs and 8 drainages procedures were selected, and 22 patients were men and 25 women. The mean age of patients was 61 (range 18–93) (Table 1).

All information regarding human material was managed using anonymous numerical codes, and all samples were handled in compliance with the Helsinki Declaration (http://www.wma.net/en/30publications/10policies/b3/). This was a retrospective study and no extra tests were conducted on the patients’ material.

### 2.1. US-FNA Procedure

US-FNA was performed using a fine needle which can be visualized during the aspiration procedure and that is constantly monitored while it is guided into the target lesion (identified by Computed Tomography (CT) scan, Figure 1A) with the use of a dedicated probe with a central hole (Figure 1B) through which the needle set is introduced (Figure 1E). This is still the most suitable and reliable approach for these purposes because the needle is visible in real time during the aspiration [14]. The Menghini-type semiautomatic device was used (BIOMOL, HS, Roma, Italy), which allowed vacuum aspiration with a 20 G needle (Figure 1C) for CB and smear preparation (Figure 1D). The cytological sample was promptly inserted into a container filled with calcium chloride and plasma solution at a 1:1 concentration. The residual sample was directly smeared on the slide and fixed into an alcohol 95° solution.

### 2.2. EBUS-FNA Procedure

EBUS-FNA was performed using an ultrasound bronchoscope (BF-UC180F; Olympus, Tokyo, Japan) in combination with an ultrasound processor (EU-ME2; Olympus) under local anesthesia and conscious sedation. The ultrasound bronchoscope was passed through the mouth to the trachea (Figure 2A); then the ultrasound transducer was moved around in search of possible lesions on the airway wall. Once target lesions were visualized by ultrasound imaging, needle aspirations with a 22 G needle were performed under real-time ultrasound guidance (NA-201SX-4022; Olympus) (Figure 2B). The aspirated specimen within the needle was promptly pushed out by a needle stylet into a container filled with a 10% neutral buffered formalin solution. EBUS-FNA was performed with at least 4 passes for each target.

### 2.3. Drainage Procedure

Thoracentesis and paracentesis were performed with the introduction of a needle into the thoracic or abdominal cavity to extract the liquid content; the needle was connected to a small tube through which it was possible to collect the fluid.

### 2.4. Complications of Procedures

There were no complications after EBUS-FNA, US-FNA and drainage procedures.

### 2.5. CC Procedure

The CC procedure consisted of cytological samples processed by smear and cytocentrifuge, according to the agreed international procedure.

Specifically, smears were prepared directly by the clinician immediately after US-FNA and placed in an 95° alcohol fixative solution. Finally, they were stained with the Papanicolau method inside the laboratory.

For drainage, the cytocentrifuge method was used. In detail, drainage samples were received in a 50% alcoholic solution and precentrifugated in falcons at 2500 rpm for 5 min; after the elimination of the supernatant, the cell button was resuspended, vortexed and put in four 0.5 mL/cuvettes for each sample for a second cytocentrifugation at 800 rpm for 4 min. Finally, re-fixation with cytofix for 60 min was performed, and the four cellular spots were stained with May–Grünwald–Giemsa, Papanicolau and hematoxylin and eosin (H&E) (2 sections) stains, respectively.

### 2.6. CB Procedure

Samples obtained from EBUS-FNA and fixed in a 10% neutral buffered formalin solution (Figure 3A) were automatically processed as histological biopsies and embedded in paraffin (Figure 3D). Samples obtained from US and received in calcium chloride and plasma solution were allowed to settle until clot formation (Figure 3B) with a subsequent fixation in 10% buffered formalin. Finally, the solid clot was automatically processed (Figure 3C) and embedded in paraffin (Figure 3D). Four micron-thick sections were stained with hematoxylin and eosin (H&E) and analyzed for adequacy. Immunocytochemistry was subsequently performed on adequate samples if malignancy was suspected. In 4 cases, received after 2016, FISH analysis was carried out to subtype the lymphoma according to the 2017 WHO classification.

Samples obtained from drainages were centrifuged at 2500 rpm for 5 min. If the residual cellularity was scant, samples were placed in a calcium chloride/plasma solution; if the residual cellularity was in a greater amount, samples were fixated in 10% buffered formalin.

To avoid preanalytic artifacts that could affect morphological interpretation, immunocytochemistry and FISH analyses, an internal agreement among pathologists, technicians and interventional operators has been made on the timing of sample collections and sample deliveries to the pathology laboratory. In particular, all the FNAs were conducted in the morning and taken to the laboratory in the early afternoon; the samples were processed in the same evening when they were obtained, in a time-fixation range of 6–12 h. This caution avoids the hyperfixation of the sample.

Our internal protocol of CB preparation was improved in March 2014 with the help of an expert cytopreparatory technician (GB).

### 2.7. Immunocytochemical Analysis

Immunocytochemical analysis was possible only on CBs, on three micron-thick unstained sections. An initial panel of antibodies was used in malignant cases to differentiate lymphoma from carcinoma and sarcoma, namely pan-cytokeratin (clone AE1/3; DAKO, Carpinteria, CA, USA), CD45/LCA (Clones 2B11+PD7/26; DAKO), CD20 (clone L26; DAKO), CD79α (clone JCB 117; DAKO), CD3 (polyclonal; DAKO), EMA (clone E29; DAKO), CD30 (clone Ber-H2; DAKO), CD15 (clone Carb-3; DAKO) and PAX5 (clone DAK-PAX5; DAKO). In case a diagnosis of B-lineage NHL had been advanced, a second lineage of immunocytochemical markers was performed, including BCL2 (clone 124; DAKO), BCL6 (clone PG-B6p; DAKO), CD10 (clone 56C6; DAKO), CD5 (clone 4C7; DAKO), CyclinD1 (clone EP12; DAKO), MUM1/IRF4 (clone MUM1p; DAKO) and Mib1 (clone Ki67; DAKO). All staining procedures were carried out on Autostainer Link48 (DAKO, DAKO), according to the manufacturer’s instructions and in the presence of appropriate controls. Neither morphology nor first- nor second-line immunocytochemical analyses aroused suspicion of T-lineage NHL.

### 2.8. FISH Procedure

FISH was needed to subclassify 4 cases of NHL according to the WHO 2017 classification. In two cases, to establish an unequivocal diagnosis of Mantle Cell lymphoma (MCL) (cases 20 and 47), FISH was performed, according to the manufacturer’s instructions (Abbott, Chicago, IL, USA), using Vysis LSI IGH/CCND1 XT Dual Color Dual Fusion Probes, a pool of probes hybridizing to chromosome 14q32 (IGH Spectrum Green) and chromosome 11q13 (CCND1 Spectrum Orange) (Abbott, Chicago, IL, USA), which is able to detect the translocation t(11;14)(q13; q32). The hybridized slides were viewed on an Olympus IX-50 epifluorescence microscope (Olympus) equipped with a cooled CCD camera (Princeton Instruments) at 100× magnification with oil immersion. The GenASIs software (Applied Spectral Imaging, Carlsbad, CA, USA) was used to generate multicolor images. t(11;14) translocation was assessed by scoring all evaluable *nuclei*. In normal intact cells, two separate red and two separate green individual signals were visible, whereas a reciprocal translocation t(11;14)(q13; q32) generated two fused red/green signals accompanied by one red and one green signal (representing the normal loci). To investigate a possible diagnosis of high-grade lymphoma with double or triple translocations (HG lymphoma Double Hit/Triple Hit according to 2017 WHO) in two cases (cases 21 and 33), FISH analysis using MYC (Vysis LSI MYC Dual Color Break Apart Rearrangement Probe; Abbott, Chicago, IL, USA), BCL2 (Vysis LSI BCL2 Dual Color, Break Apart FISH Probe; Abbott, Chicago, IL, USA), and BCL6 (Vysis LSI BCL6 Dual Color Break Apart Rearrangement Probe; Abbott, Chicago, IL, USA), were performed according to the manufacturer’s instructions. Each assay is a pool of probes hybridizing upstream (green probes) and downstream (red) of the breakpoint of the gene. The MYC, BCL2, and BCL6 normal signals are a yellow signal from fused red and green fluorescence. FISH pattern hybridization was analyzed according to the manufacturer’s instructions.

## 3. Results

### 3.1. Cytological Samples and Disease Location

Forty-seven patients were identified as follows: for 20 patients, only CC (smears or cytocentrifugate) was performed; for 15 patients, only the CB procedure was carried out; in 12 patients, the specimens were treated to obtain both CCs and CBs. In these 12 patients, a definitive diagnosis of lymphoma could be performed on CBs. Targeted lesions were distributed as follows: mediastinum (10), pleural/peritoneal effusions, (8), deep soft tissues (10), deep lymph-nodes (11), liver (2), lung (2) and spleen (4) (Table 1).

### 3.2. Cytological Diagnosis and Ancillary Studies

Of 47 specimens, 33 (70%) were adequate for a diagnosis of lymphoma, in particular 12 (60%) of the 20 CCs and 21 (78%) of the 27 CBs (Figure 4).

For samples analyzed only by CC, only the diagnosis of lymphoma could be suggested, and the need for integration with clinical data and CF was underlined in the cytological report. Eight out of 47 cases were inadequate for a definitive diagnosis because of scant cellularity: in particular, 5 (25%) of the 20 CCs and 3 (11%) of the 27 CBs. Six cases were diagnosed as suspicious for lymphoma, specifically 3 (15%) of the 20 CCs and 3 (10%) of the 27 CBs. Within CB samples, 5 were morphologically compatible with HL, classical type; therefore, immunocytochemistry was performed. In three cases, immunocytochemistry confirmed the diagnosis: large mononucleated cells and Reed–Sternberg cells were strongly positive for CD30 and CD15, faintly positive for PAX5 and negative for CD45, CD3 and CD20 (cases 19, 24, 25; Figure 5, case 25).

In the remaining two cases, immunocytochemistry failed to confirm a diagnosis of HL because of low cellularity but a strong suspicion of HL, classical type, was advanced (cases 2, 30) and, in one case, the diagnosis was confirmed from a subsequent biopsy (case 30). No conclusive diagnosis of HL was made on CC samples. In 16/21 CBs, we were able to perform immunocytochemistry to subclassify NHL according to the 2017 WHO classification; in four cases, FISH was also performed and allowed to confirm MCL (cases 20, 47; Figure 6, case 20) and to classify diffuse large B-cell lymphoma as post germinal center type (case 21), respectively. In case 33, FISH was not conclusive for poor cellularity but a diagnosis of DLBCL not otherwise specified (NOS) was made on the basis of the morphological and immunocytochemical findings only. We performed FISH analyses for c-Myc, BCL2 and BCL6 genes to subclassify diffuse large B-cell lymphoma on cases received after the 2017 WHO edition.

Interestingly, among CB cases, one patient presented with clinical and radiological findings (pulmonary cystic lesions) which were tuberculosis-suspicious, but the final diagnosis was HL, classical type, as previously reported [15]. Fifteen patients’ samples were sent to our attention with a clinical hypothesis of carcinoma. Of these, in 13 cases CBs were obtained and in 9/13 cases immunocytochemistry was conducted to confirm lymphoma: two HL and seven NHL, respectively. One case was strongly suggestive of HL, but the diagnosis could not be confirmed by immunocytochemistry because of low cellularity. With the last case being a follow-up of a previous HL, classical type, a diagnosis of “strongly suggestive for HL” was sufficient for the clinician.

Therefore, in nine cases, surgery for carcinoma was ruled out and further hematological investigations were scheduled.

Of five patients with a final diagnosis of HL, classical type, on CBs, three cases received a diagnosis of lymphoma that was not clinically advanced, and two of these patients were suspected to have carcinoma and one patient suspected to have tuberculosis on the basis of the radiological findings. In the remaining two cases presenting with mediastinic mass, an epithelial neoplasm arising in the follow-up for HL and a differential diagnosis between lymphoma and thymoma were advanced, respectively.

### 3.3. Statistical Analysis

The two different methods (CCs and CBs) were compared using the Pearson Chi-square test or Fisher’s exact test, when indicated, and *p*-values < 0.05 were considered statistically significant.

The percentage of “compatible” results observed in CBs method (78%) was higher than in CCs method (60%) but not statistically significant (*p* = 0.187) when we compared the “compatible” with “inadequate” and “suspicious” results (Table 2).

However, a “compatible” diagnosis of NHL or HL on CB was definitive, while a “compatible” diagnosis on CC needed CF to confirm the diagnosis of NHL.

## 4. Discussion

In order to make a diagnosis of lymphoma and its subclassification to consequently plan the most appropriate and effective therapy, clinical data, morphology, immunophenotype and molecular analyses are fundamental [2,16]. Although a histological evaluation of surgically excised specimens remains the gold standard for the correct diagnosis of lymphoma and classification, in selected cases of deep-seated lesions, extra-nodal sites, recurrent disease and patients with poor performance status, a less-invasive procedure might be considered as an option to obtain diagnostic samples [1,3,7,8,9]. EBUS-FNA and US-FNA are minimally invasive and safe techniques with low complications rates in obtaining tissue samples from deep-seated lesions and extra-nodal sites, such as mediastinal, retroperitoneal and deep-seated lymph-nodes or masses [7]. With US-FNA and EBUS-FNA, it is possible to obtain cytological material, which is either smeared on glasses or analyzed after cytocentrifugation [14]. This procedure is simple, fast, cheap, largely diffuse and used daily in pathology laboratories. On the other hand, there are some general limitations in the use of CC for a diagnosis of lymphoma which impact on sensitivity to some extent [1]. In particular, smear suffers from artifacts derived from poor fixation, preparation, overcrowding of cells or staining techniques; cytocentrifugate lacks tissue architecture, and is affected by low neoplastic cellularity which does not allow immunocytochemical studies nor molecular analysis and/or FISH [13]. For all these reasons, CC by itself is commonly inadequate and a diagnosis of NHL on CC needs integration with CF [13]. With the aid of CF, the sensitivity, specificity and accuracy of a diagnosis of NHL increase to 74%, 93% and 81%, respectively, both in US-FNA [1,8] and in effusion cytology [17]. However, CF has limited utility in large B-cell lymphoma and in T-cell lymphoma needing T-cell receptor γ gene rearrangement clonality [7,9,10].

Another well-recognized limitation of CC is the difficulty to diagnose HL because of the paucity of neoplastic cells [18]. In HL, the physician needs to provide the pathologist with an adequate diagnostic specimen [19]. HL has the unique feature of malignant cells representing a minority of the population within a rich environment composed of reactive lymphocytes, eosinophils, and histiocytes [2]; therefore, smears may not include sufficient malignant cells or may be constituted only by reactive cells [18]. Moreover, if the classic HL morphology is observed on smears, immunocytochemistry cannot be performed to confirm the nature of the atypical cells, such as Reed–Sternberg cells in classical HL or Hodgkin’s cells in nodular lymphocyte-predominant HL (NLPHL), and CF is neither necessary nor useful [7,9,18]. Moreover, even when tissue is easily obtained in HL patients, the necrosis or paucity of neoplastic cells may, again, preclude the correct diagnosis [8]. For this reason, FNA or a core biopsy are considered inadequate in case of suspicion of HL [19].

All this reinforces the need for more invasive procedures to obtain adequate samples for morphological, immunocytochemical and molecular analyses, in order to perform the correct diagnosis and subclassify NHL and HL. On the other hand, in deep-seated lesions or masses, in patients in which a surgical procedure and/or large-core needle biopsy could be risky in terms of potential injury to vital structures, in cystic lesions (as some cases of Primary Pulmonary Hodgkin lymphoma) and in extra-nodal visceral localization of lymphoma, FNA is the only feasible procedure [20].

The aim of CB is to better preserve tissue architecture and morphological details to distinguish malignant from non-malignant cells as well as to obtain adequate material for immunocytochemical and molecular analyses [21]. The advantages of CB with a standardized procedure include: reduced cellular dispersal, which allows microscopic evaluation more easily than in traditional smears; recognition of disease patterns which are not always reliably identified on conventional smears; possibility to study multiple sections by routine and special stainings, immunocytological procedures and molecular analysis as is well established in advanced pulmonary carcinoma [11,12]. Finally, the CB method allows the storing of slides for retrospective studies [11,12,19].

In our retrospective series, we were able to make a definitive diagnosis of lymphoma in 21/27 CB cases (78%) compared to 12/20 CC cases (60%). In CB samples, we were able to perform a definitive diagnosis of HL, classical type, in three cases; a strongly suspected diagnosis of HL, classical type, in two cases, and, in 16 NHL cases, to carry out ancillary studies to subclassify them according to the WHO classification. Conversely, in CC samples, the diagnosis of NHL needed integration with CF and no diagnosis of HL was possible. In about half of CB cases, lymphoma was not clinically suspected and CB allowed us to perform ancillary studies and guide clinical management.

Regarding the diagnosis of HL on cytological samples, very few studies have been described in the literature.

The diagnostic efficacy of CC in HL diagnosis was described in a series of patients with a proven histopathological diagnosis of classical HL [22] and classical HL, nodular sclerosis variant (NSCHL) [23], respectively. Smears from these patients were retrospectively collected to detect HL findings, but immunocytochemistry could not be performed to confirm neoplastic cells. Therefore, the authors concluded that histopathological confirmation of surgical samples is essential to make a diagnosis of HL [22,23].

Recently, the limited sensitivity of CB in a retrospective series of patients with superficial nodal localization of NLPHL variant has been described [24]. Despite the well-established and recognized expertise in the hematopathological field of the first author and the easy acquisition of cytological samples from superficial nodes, the very rare neoplastic components of NLPHL (<1% of the total cellularity) and the challenging differential diagnosis with its mimics, including non-lymphomatous disease and some of NHL, can significantly affect CB utility in this setting, as described [25].

Nevertheless, it is imperative to look for alternative methods to surgical biopsies in “selected clinical practice” and improve our diagnostic ability, trying to better deal with available cytological samples.

In our series, CB was very useful in establishing the first diagnosis of HL, especially in patients whose clinical presentation and radiological findings were not predictive of a diagnosis of lymphoma.

In detail, case number 19 was a 69-year-old woman presenting with asthenia and fever. A CT scan showed multiple, confluent mediastinal lymphoadenopathies, up to 9 cm in maximum diameter, which is suspicious in terms of carcinoma. CB was obtained from EBUS-FNA and a HL diagnosis was possible with the aid of immunocytochemistry. Because of the strong clinical suspicion of carcinoma, a mediastinal biopsy was immediately performed, and a diagnosis of classic HL was confirmed.

Case number 24, previously described [15], was an 18-year-old woman with no previous oncological history, who presented to our Institute with a clinical-radiological diagnosis of tuberculosis. An enhanced chest CT scan showed large, partially cavitated lesions, 8 cm in maximum diameter, involving the right lung, without mediastinal lymph node enlargement. After an initial improvement of the patient’s clinical condition due to steroid treatment, a worsening of symptomatology occurred and a pulmonary biopsy was programmed. US-FNA was the only feasible procedure on a cystic pulmonary lesion. HL diagnosis on CB was made, supported by the immunocytochemical findings. After an appropriate therapy for HL, the patient is alive and in an excellent health condition.

Case number 25 is a 41-year-old man diagnosed with carcinosarcoma of the glands. Synchronously, the CT scan documented an intraosseous pelvic mass and confluent lymphoadenopathies with a diameter greater than 11 cm located in the mediastinum, extending to pulmonary hilum and supraclavear region and involving large caliber vessels. The clinical-radiological hypothesis was metastases from the penile carcinosarcoma. CB was obtained from EBUS-FNA on mediastinal mass, and a diagnosis of HL was made. After appropriate therapy for HL and penile carcinosarcoma, the patient is alive and in a good health condition.

Case number 30 is a 30-year-old man, previously affected by adult-onset Still’s disease, presenting with asthenia and fever. The CT scan showed an antero-superior mediastinal mass of 9 cm in maximum diameter, clinically suspicious for thymoma/lymphoma. US-FNA was performed and CC and CBs were obtained. Based on morphology (immunocytochemistry was not useful because of low cellularity), the findings were strongly suspicious for HL. Surgery for thymoma was not performed, and a mediastinal biopsy was required, confirming the diagnosis of HL.

As can be guessed, these four patients were not suitable for surgical procedure, and FNA provided cytological samples which CB made adequate for the diagnosis of HL. These unexpected diagnoses allowed the clinicians to choose the best medical management for the patients and, in case 30, to switch the therapy from surgical to medical.

Finally, case number 2 was a 19-year-old woman with a diagnosis of nodal classic HL from 2014. After disease progression, with evidence of mediastinal and abdominal lymphadenopathies and multiple nodularities in the pulmonary parenchyma, US-FNA of the mediastinal lymph nodes was performed. A diagnosis of strong suspicion of HL on CBs was made, although immunocytochemistry was not informative and our diagnosis was sufficient to schedule an adequate therapy.

We are aware that the retrospective nature of this study is a major limitation and that the small number of cases does not allow an adequate statistical analysis. However, we would suggest that in selected patients, CB represents a valuable diagnostic tool to choose the most appropriate clinical management and therapeutic approach, also in the diagnosis of HL.

Moreover, CB obtains an adequate cellularity for the diagnosis of lymphoma also from drainage, which is well known to provide samples with a poor cellularity.

## 5. Conclusions

A correct diagnosis and subtyping of lymphoma is imperative for adequate therapy and clinical management. For deep-seated and extra-nodal lesions, mostly in patients with comorbidities or unfitness, an easy and less invasive method, such as US-FNA and EBUS-FNA, is needed for the collection of samples. CB is a very useful method to perform a morphological analysis, as well as immunocytochemical and molecular studies. In our study, we confirm that CB is more useful than CC in achieving a diagnosis of NHL and its subclassifications, and we demonstrate CB utility in the first diagnosis of HL, classical type, in patients with deep-seated lesions or unusual presentation.

## Figures and Tables

**Figure 1 diagnostics-10-00748-f001:**
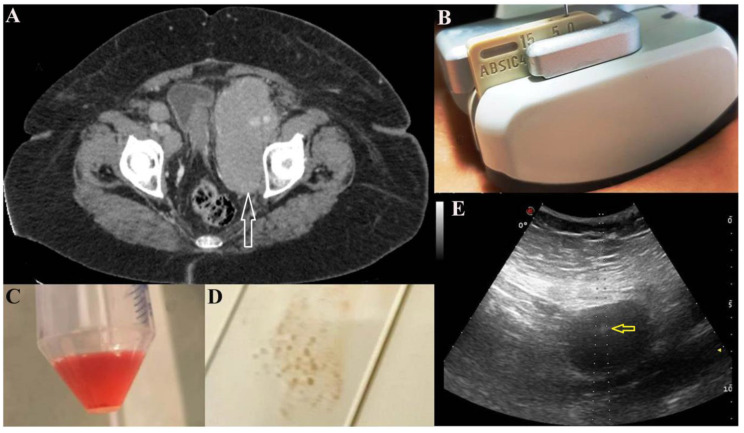
Computed tomography (CT) scan: solid abdominal/retroperitoneal pelvic mass (white arrow) (**A**). US-FNA: a dedicated probe with a central hole through which the needle set is introduced (at angle: 0°) (**B**). Cytological sample aspirated for CB (**C**) and smear for conventional cytology (CC) (**D**). Needle (yellow arrow) within the mass (**E**).

**Figure 2 diagnostics-10-00748-f002:**
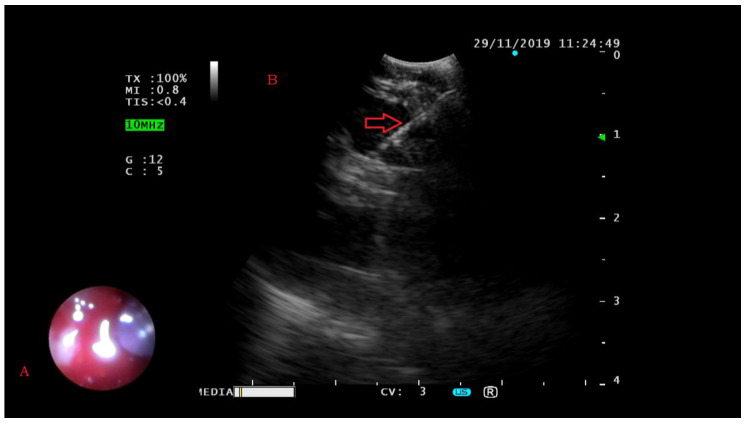
Trachea view by endoscopy (**A**). Introduction of fine needle into target lesions (red arrow) under real-time EBUS (**B**).

**Figure 3 diagnostics-10-00748-f003:**
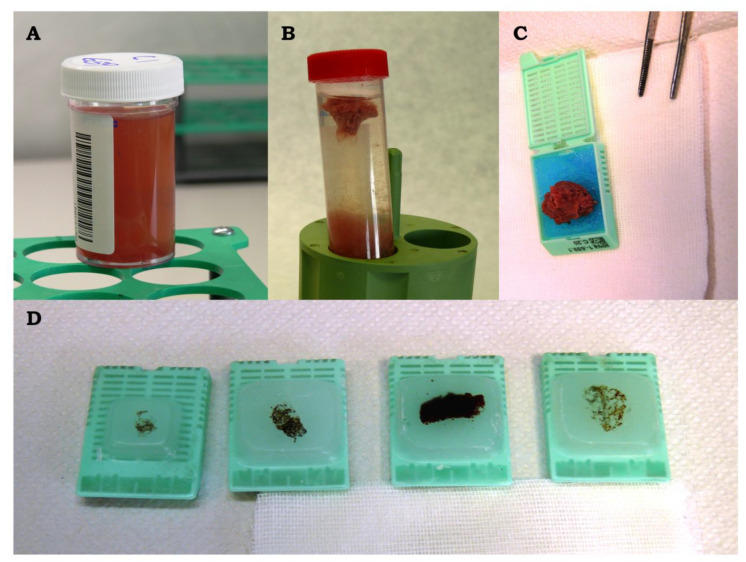
Cytological specimens into container with 10% neutral buffered formalin solution (**A**); solid clot (**B**) into cell for processing (**C**) and embedded in paraffin (**D**; 4 samples).

**Figure 4 diagnostics-10-00748-f004:**
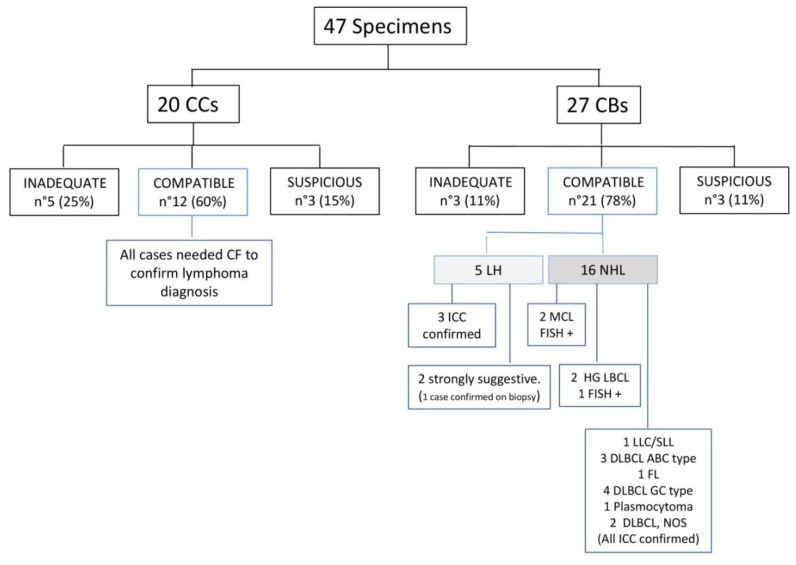
Diagnosis from CCs and CBs. CB: Cell Block; CC: Conventional Cytology; CLL/SLL: Chronic Lymphocytic Leukemia/Small Lymphocytic lymphoma; DLBCL, ABC-type: Diffuse Large B-cell lymphoma, Activated B-cell type; DLBCL, GCB type: Diffuse Large B-cell lymphoma, Germinal Center type; DLBCL, NOS: Diffuse Large B-cell lymphoma, Not Otherwise Specified; FISH: Fluorescence In Situ Hybridization; FL: Follicular lymphoma; HL: Hodgkin lymphoma, MCL: Mantle Cell lymphoma.

**Figure 5 diagnostics-10-00748-f005:**
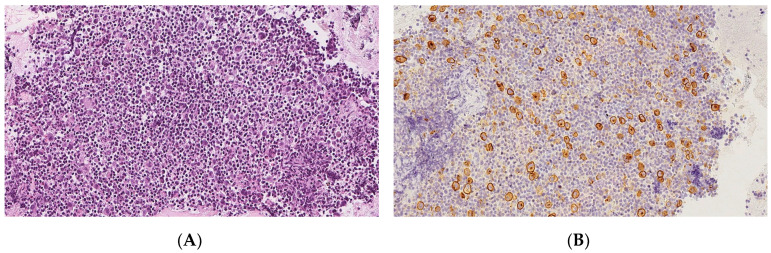
Hodgkin Lymphoma, Classic Type, case 25. Hematoxylin–Eosin CB-stained section shows small lymphocytes, eosinophils and histiocytes mixed to scattered large mononucleated and multinucleated cells (30×) (**A**). CD30 highlights large atypical cells (30×) (**B**).

**Figure 6 diagnostics-10-00748-f006:**
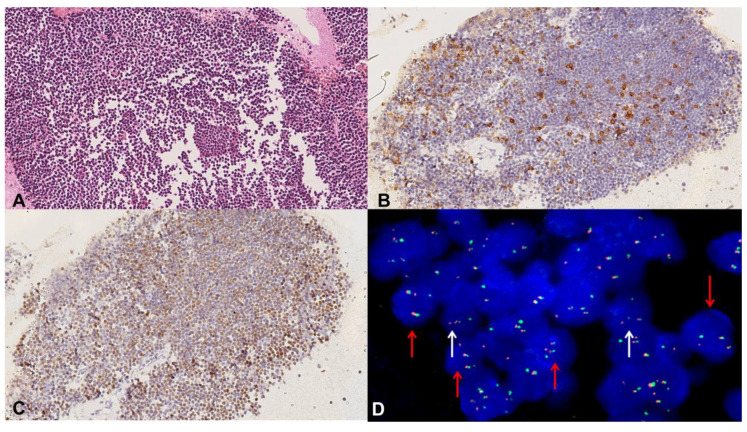
Mantle Cell lymphoma, case 20. Hematoxylin–Eosin CB-stained section shows small monomorphic lymphocytes (20×) (**A**). CD5 stains small lymphocytes (20×) (**B**). CyclinD1 expression in neoplastic cells (20×) (**C**). FISH analysis showing the t(11;14)(q13; q32) translocation. The red arrow indicates neoplastic cells containing single red and green signals (corresponding to the normal CCND1 and IGH loci, respectively) together with two fused signals (corresponding to the genes fused by the translocation). Normal cells are indicated by the white arrow (**D**).

**Table 1 diagnostics-10-00748-t001:** Patient demographics, clinical findings, procedure details and diagnosis. CB: cell block; Cc: Cytocentrifugate; CLL/SLL: Chronic Lymphocytic Leukemia/Small Lymphocytic lymphoma; DLBCL, ABC-type: Diffuse Large B-cell lymphoma, Activated B-cell type; DLBCL, GCB type: Diffuse Large B-cell lymphoma, Germinal Center type; DLBCL, NOS: Diffuse Large B-cell lymphoma Not Otherwise Specified; EBUS-FNA: Endobronchial Ultrasound-guided Fine Needle Aspiration; FISH: Fluorescence In Situ Hybridization; FL: Follicular lymphoma; HCL: Hairy Cell Leukemia; HL: Hodgkin lymphoma; ICC: Immunocytochemistry; LN: Lymph-node; MCL: Mantle Cell lymphoma; MM: Mediastinum Mass; NHL: non-Hodgkin lymphoma; PLE: Pleura; Sm: Smear; SP: Spleen; ST: Soft Tissue; TB: *Tubercolosis*; US-FNA: Ultrasound-guided Fine Needle Aspiration.

Case Number	Sex	Age at Diagnosis	Procedure	Site	Method	Clinical Indications	Clinical Queries	Cytological Diagnosis	ICC	FISH
1	M	75	Drainage	PLE	CB	Pleural effusion	Carcinoma	LLC/SLL	Yes	
2	F	19	US-FNA	MM	CB	Follow-up in HL	Carcinoma	Strongly suggestive for HL	Yes	
3	M	79	US-FNA	LN	CB + Sm	Soft tissue mass	Lymphoma	DLBCL, ABC type	Yes	
4	M	62	US-FNA	ST	Sm	Follow-up in NHL	Lymphoma	Compatible with NHL B	Inadequate	
5	F	78	Drainage	PE	Sm	Retroperitoneal mass	Carcinoma	Compatible with NHL B	Inadequate	
6	M	74	US-FNA	SP	Sm	MD	Lymphoma	Compatible with NHL B	Inadequate	
7	M	78	Drainage	PLE	Cc	NHL	Lymphoma	Compatible with NHL B	Inadequate	
8	M	45	US-FNA	SP	Sm	Suspicious lymphoma	Lymphoma	Compatible with NHL B	Inadequate	
9	M	33	US-FNA	ST	Sm	MM	Lymphoma	Compatible with lymphoma	Inadequate	
10	M	52	US-FNA	SP	Sm	Suspicious HCL	Lymphoma	Compatible with NHL B	Inadequate	
11	F	80	US-FNA	SP	Sm	Splenic mass	Lymphoma	Compatible with lymphoma	Inadequate	
12	F	29	US-FNA	Lung	Sm	Lung consolidation	Lobar pneumonia	Compatible with lymphoma	Inadequate	
13	F	89	US-FNA	ST	Sm	Localized lymphadenopathy	Carcinoma	Compatible with NHL B	Inadequate	
14	F	48	Drainage	PLE	Cc	Follow-up in NHL	Lymphoma	Compatible with NHL T	Yes	
15	F	55	US-FNA	ST	Sm	Lymphadenomegaly	NQ	Inadequate	Inadequate	
16	F	70	US-FNA	LN	CB + Sm	Follow-up in NHL	Lymphoma	Suspicious for lymphoma	Yes	
17	F	83	US-FNA	MM	CB	Follow-up in NHL	Lymphoma	DLBCL, GC type	Yes	
18	M	82	US-FNA	LN	CB + Sm	Retroperitoneal mass	Lymphoma	NHL B, FL	Yes	
19	F	68	EBUS-FNA	MM	CB	Suspicious carcinoma	Carcinoma	HL	Yes	
20	M	74	US-FNA	ST	CB	Suspicious carcinoma	Carcinoma	MCL	Yes	Yes
21	M	74	Drainage	PLE	CB	Suspicious carcinoma	Carcinoma	DLBCL, ABC type	Yes	Yes
22	M	74	US-FNA	Liver	CB	Suspicious carcinoma	Carcinoma	DLBCL, ABC type	Yes	
23	F	70	US-FNA	ST	CB	Bone and pulmonary lesion	Carcinoma	Plasmocytoma	Yes	
24	F	18	US-FNA	Lung	CB	Cystic pulmonary lesion	TB	HL	Yes	
25	M	41	EBUS-FNA	MM	CB	Penile carcinoma and MM	Carcinoma	HL	Yes	
26	F	65	US-FNA	MM	CB	Mediastinic mass	Carcinoma	Suspicious for lymphoma	Inadequate	
27	M	72	US-FNA	MM	CB + Sm	Suspicious lymphoma	Lymphoma	DLBCL, ABC type	Yes	
28	M	26	US-FNA	LN	Sm	Lymphadenomegaly	Lymphoma	Suspicious for lymphoma	Inadequate	
29	F	54	US-FNA	LN	CB + Sm	Follow-up in NHL	Lymphoma	DLBCL, GC type	Yes	
30	M	30	US-FNA	MM	CB + Sm	Mediastinic mass	Thymoma/lymphoma	Strongly suggestive for HL	Yes	
31	M	78	Drainage	PE	Cc	Suspicious lymphoma	Lymphoma	Compatible with NHL	Inadequate	
32	M	42	US-FNA	Liver	CB + Sm	Suspicious lymphoma	Lymphoma	DLBCL, NOS	Yes	
33	F	75	Drainage	PE	CB + Cc	Peritoneal effusion	Lymphoma	DLBCL, NOS	Yes	Yes
34	F	57	US-FNA	LN	CB	Lymphadenomegaly	Carcinoma	DLBCL, GC type	Yes	
35	F	88	Drainage	PLE	Sm	Follow-up in HL	Lymphoma	Inadequate	Inadequate	
36	F	77	US-FNA	ST	CB	Abdominal mass	Lymphoma	DLBCL, GC type	Yes	
37	F	93	US-FNA	ST	CB + Sm	Soft tissue mass	Carcinoma	DLBCL, NOS	Yes	
38	M	73	EBUS-FNA	LN	CB	Pulmonary mass	Carcinoma	Suspicious for lymphoma	Inadequate	
39	M	35	US-FNA	ST	CB + Sm	Acute leukemia	Carcinoma	Inadequate	Inadequate	
40	F	55	US-FNA	MM	CB + Sm	Suspicious lymphoma	Lymphoma	Inadequate	Inadequate	
41	F	55	US-FNA	MM	Sm	Suspicious lymphoma	Lymphoma	Suspicious for lymphoma	Inadequate	
42	F	67	US-FNA	LN	Sm	Suspicious lymphoma	Lymphoma	Suspicious for lymphoma	Inadequate	
43	F	40	US-FNA	LN	Sm	Suspicious lymphoma	Lymphoma	Inadequate	Inadequate	
44	F	48	US-FNA	LN	Sm	Suspicious lymphoma	Lymphoma	Inadequate	Inadequate	
45	F	76	US-FNA	MM	CB + Sm	Suspicious lymphoma	Lymphoma	Inadequate	Inadequate	
46	M	64	US-FNA	LN	Sm	Suspicious lymphoma	Lymphoma	Inadequate	Inadequate	
47	M	68	US-FNA	ST	CB	Follow-up in NHL-mantle cell	Lymphoma	MCL	Yes	Yes

**Table 2 diagnostics-10-00748-t002:** Statistical comparison between CCs and CBs.

Results	CCs		CBs		*p*-Value	Test Used and Result
	*n* = 20		*n* = 27			
	*n*	%	*n*	%		
Inadequate	5	25	3	11	*p* = 0.394	df2 overall (Fisher Exact Test)
Compatibile	12	60	21	78	*p* = 0.187	df1 Compatibile vs. Inadeguate&Suspicious (Pearson Chi-square Test)
Suspicious	3	15	3	11	*p* = 0.258	df1 Inadeguate vs. Compatibile&Suspicious (Fisher Exact Test)

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
