# Peer review of "Diagnosis of Hodgkin Lymphoma from Cell Block: A Reliable and Helpful Tool in “Selected” Diagnostic Practice"

_diagnostics, 2020, doi:10.3390/diagnostics10100748_

Round 1
Reviewer 1 Report
A good job based on a practical reality that determine high level results.
for this reason I recommend a comment on how to organize and how to solve the practical problems in applying such methods.
question: how were the problems of the pre-analytical phase faced and solved?
Author Response
Response to Reviewer 1 Comments
Review 1: ‘A good job based on a practical reality that determine high level results.
For this reason I recommend a comment on how to organize (1) and how to solve the practical problems in applying such methods (2).
Question: how were the problems of the pre-analytical phase faced and solved (3)?’
We are grateful to Review 1 for her/his kind considerations about our paper and for her/his useful suggestions. The paper has been reviewed as follows:
Point 1: We have specified how we have organized our ‘internal agreement between pathologists, technicians and interventional operators’ at line 165-170 and that our internal CB protocol was improved with the aid of an expert cytopreparatory technician at line 171-172 (in red).
Point 2: We have specified all the steps of CB preparation, specifically for US-FNA samples at line 109-112, for EBUS-FNA samples at line 150-153, and for drainages at line 162-164 (in red).
Point 3: We have specified how we have avoided pre-analytical artifacts at line 168-170 (in red).
We hope the required integrations may be deemed satisfactory and exhaustive.
Reviewer 2 Report
General (reviewer’s comment and opinion).
The study concerns the use of Cell Block methodology for the diagnosis of Hodgkin Lymphoma (HL) and Non-Hodgkin Lymphoma (NHL) in selected cases of deep-seated lesions or in which standard surgical procedures cannot be performed. Forty-seven cytological samples were selected: 20 for conventional cytology (CC), 15 for cell block (CB) procedure; 12 for both CC and CB procedures. Furthermore, immunocytochemical analysis and FISH procedures were performed on CB samples. For CC samples, all cases needed cytofluorimetry to confirm lymphoma diagnosis. For CB samples, 5 cases were morphologically compatible with HL, the others needed immunocytochemical and FISH procedures to subclassify NHLs.
The authors concluded that CB methodology is more useful than CC in HL diagnosis and NHL diagnosis and subclassification in patients with deep-seated lesions or unsuitable for surgical procedures.
The objective of the paper is clearly stated. However, the manuscript needs a few minor revisions.
The text is well written and easy to read with a few grammar errors that should be corrected by an English mother-tongue translator. The methods are adequately described except for the “CC procedure” section that needs more expansion. The discussion is articulated but should be shortened. A statistical analysis should be attempted even the limited number of cases may decrease the significance of the results, as already stated by the authors.
Author Response
Response to Reviewer 2 Comments
Review 2: ‘The study concerns the use of Cell Block methodology for the diagnosis of Hodgkin Lymphoma (HL) and Non-Hodgkin Lymphoma (NHL) in selected cases of deep-seated lesions or in which standard surgical procedures cannot be performed. Forty-seven cytological samples were selected: 20 for conventional cytology (CC), 15 for cell block (CB) procedure; 12 for both CC and CB procedures. Furthermore, immunocytochemical analysis and FISH procedures were performed on CB samples. For CC samples, all cases needed cytofluorimetry to confirm lymphoma diagnosis. For CB samples, 5 cases were morphologically compatible with HL, the others needed immunocytochemical and FISH procedures to subclassify NHLs.
The authors concluded that CB methodology is more useful than CC in HL diagnosis and NHL diagnosis and subclassification in patients with deep-seated lesions or unsuitable for surgical procedures.
The objective of the paper is clearly stated. However, the manuscript needs a few minor revisions.
The text is well written and easy to read with a few grammar errors that should be corrected by an English mother-tongue translator (1). The methods are adequately described except for the “CC procedure” section that needs more expansion (2). The discussion is articulated but should be shortened (3). A statistical analysis should be attempted even the limited number of cases may decrease the significance of the results, as already stated by the authors (4).’
We are grateful to Review 2 for her/his kind considerations about our paper and for useful comments and suggestions.
The paper has been reviewed as follows:
Point 1: The final draft has been linguistically and stylistically revised by a mother-tongue translator and proofreader.
Point 2: We have specified all the steps for the CC preparation, specifically smear method at line 140-142 and cytocentrifuge at line 143-148 (in red).
Point 3: We have tried to simplify the Discussion (1535 words respect to 1626)
Point 4: Finally, we have added a simple statistical analysis about the utility of CB compared to CC at line 277-281 and 285-286 and a summary table (Table 3) at line 282.
We hope the required integrations may be deemed satisfactory and exhaustive.